# The Contribution of Wnt Signaling to Vascular Complications in Type 2 Diabetes Mellitus

**DOI:** 10.3390/ijms23136995

**Published:** 2022-06-23

**Authors:** Raquel Sanabria-de la Torre, Cristina García-Fontana, Sheila González-Salvatierra, Francisco Andújar-Vera, Luis Martínez-Heredia, Beatriz García-Fontana, Manuel Muñoz-Torres

**Affiliations:** 1Department of Medicine, University of Granada, 18016 Granada, Spain; raquelsanabriadlt@gmail.com (R.S.-d.l.T.); sgsalvatierra@ugr.es (S.G.-S.); luismh95@gmail.com (L.M.-H.); mmt@mamuto.es (M.M.-T.); 2Instituto de Investigación Biosanitaria de Granada (ibs.GRANADA), 18012 Granada, Spain; andujar@ibsgranada.es; 3Endocrinology and Nutrition Unit, University Hospital Clínico San Cecilio, 18016 Granada, Spain; 4CIBER de Fragilidad y Envejecimiento Saludable (CIBERFES), Instituto de Salud Carlos III, 28029 Madrid, Spain; 5Department of Computer Science and Artificial Intelligence, University of Granada, 18071 Granada, Spain; 6Andalusian Research Institute in Data Science and Computational Intelligence (DaSCI Institute), 18014 Granada, Spain

**Keywords:** cardiovascular disease, microvascular disease, macrovascular disease, type 2 diabetes mellitus, Wnt pathway

## Abstract

Vascular complications are the leading cause of morbidity and mortality among patients with type 2 diabetes mellitus (T2DM). These vascular abnormalities result in a chronic hyperglycemic state, which influences many signaling molecular pathways that initially lead to increased oxidative stress, increased inflammation, and endothelial dysfunction, leading to both microvascular and macrovascular complications. Endothelial dysfunction represents the initial stage in both types of vascular complications; it represents “mandatory damage” in the development of microvascular complications and only “introductory damage” in the development of macrovascular complications. Increasing scientific evidence has revealed an important role of the Wnt pathway in the pathophysiology of the vascular wall. It is well known that the Wnt pathway is altered in patients with T2DM. This review aims to be an update of the current literature related to the Wnt pathway molecules that are altered in patients with T2DM, which may also be the cause of damage to the vasculature. Both microvascular complications (retinopathy, nephropathy, and neuropathy) and macrovascular complications (coronary artery disease, cerebrovascular disease, and peripheral arterial disease) are analyzed. This review aims to concisely concentrate all the evidence to facilitate the view on the vascular involvement of the Wnt pathway and its components by highlighting the importance of exploring possible therapeutic strategy for patients with T2DM who develop vascular pathologies.

## 1. Wnt Signaling Pathway in the Vasculature

All organs depend on blood vessels for oxygen and nutrients. The vasculature of each organ is structurally and molecularly different. The control of organ-specific vascularization and endothelial cell differentiation is controlled by intracellular signaling pathways. Although the best known vascular signaling pathways are vascular endothelial growth factor/vascular endothelial growth factor receptor (VEGF/VEGFR) and angiopoietin/Tie2, a growing body of literature points to the Wingless-Int (Wnt) signaling pathway as an essential component in vascular development [1]. As emerged from loss- and gain-of-function experimental models, Wnt signaling might contribute to vascular development and homeostasis [2].

The Wnt signaling pathway is made up of proteins that transmit signals from the extracellular to the intracellular matrix. These components are included in three pathway types: canonical pathway, noncanonical cell polarity pathway, and noncanonical calcified pathway [3,4] (Figure 1).

The canonical pathway is the best characterized at present, and it is involved in mediating fundamental biological processes such as embryogenesis, organogenesis, and tumorigenesis [1]. Initially, it was fundamentally linked to the regulation of bone formation, but the evidence currently points to an important role at the vascular level. 

Skeletal integrity is maintained by a balance between bone resorption and bone formation, a process called bone remodeling [5]. Runx2 is a target gene of Wnt signaling, and activation of Runx2 by Wnt stimulates osteoblast differentiation and bone formation [6,7]. Expression of Runx2 begins in uncommitted stem cells, increases in osteoblast precursors, peaks in immature osteoblast, and decreases once osteoblasts mature [8]. This expression of Runx2 is modulated by canonical Wnt signaling, resulting in an inhibition of chondrocyte differentiation in early mesenchymal cells and directing the progenitors to become osteoblasts [8]. The process occurs during embryonic development when establishing the body axis and tissue and organ development, and it functions after birth in bone maintenance and repair [7,9]. In fact, agents that are known to activate the β-catenin pathway are used to accelerate bone healing. Sclerostin was shown to suppress bone formation, and romosozumab, an antisclerostin antibody, was approved for the treatment of osteoporosis in postmenopausal women at high risk of bone fractures [10].

On the other hand, the Wnt pathway has a key role for the development and maintenance of healthy vasculature [2]. Therefore, complications involving Wnt pathway disturbances may result in impaired vascularization. Dysregulation of components of the Wnt canonical pathway generate cardiovascular inflammatory damage, alter cellular plasticity, cause intracellular cholesterol accumulation, and lead to osteofibrotic responses [11]. There is accumulating evidence for a contribution of Wnt signaling pathways in atherosclerosis and vascular aging [5]. For example, vascular smooth muscle cells (VSMCs), which line the arterial wall and function to maintain blood pressure, are hypothesized to undergo a phenotypic switch into bone-forming cells during calcification [7]. The buildup of hydroxyapatite within the arterial wall or vascular calcification, is one of the greatest contributors to vascular disease [7]. Furthermore, the angiogenic activity of endothelial cells (ECs) is influenced by Wnt signaling. Endothelial dysfunction is the earliest and most fundamental pathological change in diabetes. Wnt, Frizzled (Fzd), and follistatin-related protein (FRP) genes are expressed by ECs and VSMCs; β-catenin is stabilized in the neovascular endothelium and neointimal smooth muscle. Activation of the Wnt signaling pathway can lead to vessel remodeling, while inhibition of Wnt signaling can lead to vessel regression. In fact, the activation of the Wnt/β-catenin signaling pathway by the administration of aFGF alleviates diabetic endothelial dysfunction [12]. Recombinant human aFGF would be an effective treatment of diabetic vascular complications due to its intervention in the Wnt pathway [12]. 

In addition, Wnt signaling plays an important role in the progression of heart disease, both in metabolic alterations (insulin sensitivity) and in cardiovascular remodeling and structural changes (fibrosis, sclerosis, atheroma formation, and VSMC proliferation and hypertrophy) [11]. Accordingly, the modification of components of Wnt signaling pathway is a possible therapeutic strategy to treat vasculature-related diseases. 

Agreeing with this evidence, several drugs targeting Wnt signaling have been shown to have a positive effect in the treatment of some vascular disorders, e.g., *Salvia miltiorrhiza* [13], recombinant human aFGF [12], and liraglutide [14].

Both canonical Wnt signaling and noncanonical Wnt signaling influence the phenotypic modulation of VSMCs in cardiovascular disease (CVD). In this review, we focus on the components of canonical Wnt signaling that control microvascular and macrovascular problems in patients with type 2 diabetes mellitus (T2DM).

## 2. Vascular Complications of Type 2 Diabetes Mellitus

The global prevalence of T2DM is expected to increase dramatically in the coming decades as the population grows and ages, in parallel with the increasing burden of overweight and obesity in both developed and developing countries [15]. It is predicted that the number of cases of T2DM will rise from 415 million to 642 million by 2040 [16]. Hypertension is even more common, rising in prevalence in the same countries, with a recent worldwide estimate of 1.39 billion cases [17]. Aspects of the pathophysiology are shared by both conditions, particularly those related to obesity and insulin resistance [18]. For example, in the San Antonio Heart Study, 85% of people with T2DM had hypertension by the fifth decade of life, whereas 50% of hypertensives had impaired glucose tolerance or T2DM [18]. Chronic hyperglycemia and insulin resistance play an important role in the initiation of vascular complications of diabetes [19]. Furthermore, medial vascular calcification is a predictor of cardiovascular events such as hypertension, stiffness, and even heart failure [7]. Hypertension is a major factor that contributes to the development of the vascular complications of T2DM [19]. 

Depending on whether small or large arteries are affected, microvascular complications such as retinopathy, nephropathy, and neuropathy or macrovascular complications such as coronary artery disease (CAD), cerebrovascular disease (CD), and peripheral arterial disease (PAD) may result [20] (Figure 2).

The review aims to summarize current knowledge on the involvement of the Wnt pathway in the pathobiology of macro- and microvascular complications associated with T2DM (Figure 2). Viable predictive biomarkers of vascular pathological events are reviewed, and possible strategies used to manipulate the pathway for therapeutic purposes are discussed to highlight feasible future directions through targeted and specific therapies.

## 3. Wnt Pathway and Microvascular Disease in Type 2 Diabetes Mellitus

Patients with T2DM usually present complications related to the deterioration of the vascular system which are classified as microvascular disease when it affects small vessels [21]. The microcirculation is a network of blood vessels <150 μm in diameter, comprising arterioles, capillaries, and venules [22]. This network is responsible for the primary function of the vascular tree and regulation of tissue perfusion for optimal exchange of gases and removal of metabolic waste products, and it may contribute to the unexplained variance in the association between T2DM and hypertension [21]. Small arterioles and capillaries also exhibit differential vascular remodeling in response to hypertension and T2DM [22]. The number of vessels perfused in the vascular bed and the arteriolar diameter determine the peripheral vascular resistance [23].

The main manifestations of the microvascular disease related to T2DM are diabetic retinopathy, diabetic nephropathy, and diabetic neuropathy [24] (Figure 2). The mechanisms that lead to vascular damage are multiple and involve various alterations in signaling pathways, including the Wnt pathway [13,25]. Table 1 shows some components of the Wnt pathway that are altered in the microvascular diseases in T2DM patients.

### 3.1. Retinopathy

Diabetic retinopathy is the leading cause of premature blindness among T2DM patients [22]. Mutations in Wnt signaling cause defective retinal vasculature in humans with some characteristics of the pathological vessels of retinopathy [27]. The mechanisms underlying the participation of canonical Wnt signaling in retinopathy may be associated with unbalanced oxidative stress, as well as overexpressed angiogenic and inflammatory factors [28].

Frizzled 4 and 7 (Fzd4 and Fzd7) receptors and low-density lipoprotein (LDL) receptor-related protein 5 and 6 (LRP5 LRP6) are significantly increased in pathological neovascularization in a mouse model of retinopathy [27,28] (Table 1). In a similar way, retinal β-catenin levels are increased in the cells inside the retina of patients with diabetic retinopathy compared to nondiabetic donors [26] (Table 1). 

Inhibitors of the Wnt pathway such as SERPINA3K, very LDL receptor extracellular domain (VLN), endostatin, kallistatin, pigment epithelium-derived factor (PEDF), and miR-184 are decreased in the eyes of diabetic retinopathy patients [29,30,31,32,33,34] (Table 1).

The Wnt pathway is overexpressed in retinopathy models; hence, blocking Wnt signaling with Dickkopf-1 (Dkk1) reduces retinal inflammation, as well as improves vascular leakage and neovascularization in diabetic retinopathy models [26]. Furthermore, nanoparticle-mediated VLN expression may represent a new therapeutic approach to treat pathological ocular angiogenesis and potentially other vascular diseases affected by Wnt signaling [30]. Endostatin gene transfer may provide a way to lower the risk of three causes of visual loss: macular edema, neovascularization, and retinal detachment [31]. Taken together, blocking the Wnt pathway appears to be an effective therapeutic strategy for the treatment of retinopathy.

### 3.2. Diabetic Kidney Disease

Diabetic kidney disease (DKD) is a major microvascular complication of T2DM and can lead to end-stage renal failure, in addition to contributing significantly to cardiovascular morbidity and mortality [45]. 

Despite being relatively silent in normal adult kidneys, Wnt/β-catenin signaling is reactivated in a wide range of chronic kidney diseases (CKD), such as diabetic nephropathy, obstructive nephropathy, Adriamycin nephropathy (ADR), polycystic kidney disease, and chronic allograft nephropathy [40]. Wnt/β-catenin signaling activation is one of the most relevant mechanisms of cellular senescence in diabetic nephropathy [40].

In diabetic status, the accumulated intracellular reactive oxygen species (ROS) might divert the limited pool of β-catenin from T-cell factor/lymphoid enhancer factor family (TCF/LEF) to forkhead box O (FOXO)-mediated transcription, which leads to insulin deregulation [46]. Recent evidence suggests that the renin–angiotensin–aldosterone system (RAS) is mediated by Wnt/β-catenin signaling [40]. Ectopic β-catenin causes the upregulation of all RAS genes [40]. Notably, RAS activation contributes to renal aging through various mechanisms. Several studies discovered that angiotensin II can induce senescence in renal cells [47]. Angiotensin II induces premature senescence via both signal transducer and activator of transcription 3 (STAT3) and mammalian target of rapamycin (mTOR)-regulated autophagy and the p53/p21 pathway [47], which further drives fibrosis and a redox state [48]. Furthermore, recent studies reported that RAS activation plays a role in Wnt9a-induced tubular senescence and renal fibrosis [49] (Table 1).

The causes of excess cardiovascular mortality associated with CKD have been attributed in part to the bone-mineral disorder syndrome of CKD (CKD-MBD). Wnt inhibitors, especially Dickkopf 1 (Dkk1), produced during renal repair, are involved in the pathogenesis of the vascular and bone components of CKD-MBD [39] (Table 1).

Therefore, components of Wnt pathway are potential therapeutic targets of cellular senescence in diabetic nephropathy and provide important clues for clinical strategies (Table 1). For example, liraglutide treatment ameliorates renal injury of diabetic nephropathy by enhancing Wnt/β-catenin signaling [14].

### 3.3. Neuropathy

Diabetic peripheral neuropathy (DPN) manifests as sensory symptoms in a distal “glove and a half” distribution, resulting in debilitating pain in the form of paresthesia, hyperalgesia, and allodynia. It has been reported that approximately 66% and 59% of type 1 and type 2 patients have the trend to develop different degrees of neuropathy, respectively [50,51].

Hyperglycemia, in addition to inducing oxidative stress in neurons, leads to activation of multiple biochemical pathways which constitute the major source of damage and are potential therapeutic targets in diabetic neuropathy [52]. Among them, Wnt pathway signaling could be one of those responsible for the pathogenesis and progression of diabetic neuropathy [52]. The expression of Wnt3a, β-catenin, c-myc, Dvl1, MMP2, and GRP78 is significantly increased in the sciatic nerves, spinal cord, and dorsal root ganglia of diabetic animals [43] (Table 1). Activation of the Wnt/β-catenin pathway in the spinal cord is a universal feature of the pathophysiology of neuropathic pain [44].

Inhibitors of Wnt signaling are capable of attenuating mechanical, neuropathic, and functional pain (heat, cold, and mechanical hyperalgesia), as well as improving sensory and motor nerve conduction velocity and nervous blood flow [43]. In addition, these inhibitors are able to reduce inflammation and ROS formation, as well as improve the density of peripheral nerve fibers [43,44].

A study reported that elevated spinal cord area index (SCAI) through upregulation can reduce the development of DPN by inhibiting the Wnt/β-catenin pathway upregulation [53]. This study suggested that targeting the SCAI signaling may be an alternative approach for treating DPN [53].

## 4. Wnt Pathway and Macrovascular Disease in Type 2 Diabetes Mellitus

Macrovascular complications are the leading cause of morbidity and mortality in patients with T2DM worldwide. Macrovascular disease includes CAD, CVD, and PAD [54]. At least 65% of T2DM patients die with some form of heart or CD, and the frequency of cardiovascular death in T2DM adults is 2–4 times higher than in their nondiabetic counterparts [55].

Activation of the Wnt pathway is critical for the induction of vascular injury, which regulates VSMC proliferation and apoptosis [56]. In addition, the Wnt signaling pathway plays a role in regulating inflammation in vessels, inducing endothelial cell proliferation and survival, and enhancing monocyte adhesion and trans-endothelial migration [57]. These pathways also play a key role in angiogenesis [58]. It is, therefore, important to maintain the stability of the Wnt pathway in the vessels. Table 2 shows components of the Wnt pathway that are altered during the macrovascular complications of T2DM. 

### 4.1. Coronary Artery Disease

Atherosclerosis is the main pathological mechanism in macrovascular disease, inducing an inappropriate proliferation of vascular smooth muscle cells (VSMCs), which is linked to thickening of the arterial wall, atheroma plaque formation, and vascular calcification [60,85]. This frequently results in luminal narrowing and reduced blood flow to the myocardium [85].

In T2DM patients, abnormal canonical Wnt signaling has been implicated in disturbances of the lipids, glucose, and bone homeostasis [60]. The Wnt pathway is involved in all different stages of the atherosclerotic process, from endothelial dysfunction to lipid deposition and from initial inflammation to plaque formation [59]. Sclerostin is a modulator of the Wnt/β-catenin signaling pathway in different tissues, and it has recently been linked to vascular biology [59].

Sclerostin levels are often elevated in T2DM patients [86], and sclerostin levels are positively correlated with CVD risk [60]. In fact, a cross-sectional study in a Spanish population demonstrated a positive and independent association between serum sclerostin levels and atherosclerosis in T2DM patients showing high concentrations of sclerostin associated with abnormal intima-media thickness (IMT), carotid plaques, and aortic calcifications in T2DM patients [60]. Lastly, high sclerostin levels are related to mortality due to cardiovascular causes [59]. In the same way, high serum levels of Dkk-1 are associated with acute ischemic stroke [71], and low serum levels of Wnt-1 are associated with premature myocardial infarction [87]. 

Wnt/β-catenin signaling is also essential for endothelial–mesenchymal transition, an active contributor to pathologies such as atherosclerosis and fibrosis [37]. In addition, Wnt signaling regulates lipogenesis, lipid synthesis, and ApoB secretion. In fact, administration of rmWnt3a to mice with a mutation in LRP6 (LRP6-mut) resulted in normalization of plasma LDL and triglycerides (TG) [63]. Furthermore, Wnt3a treatment of LRP6-mut mouse hepatocytes resulted in a significant reduction in hepatic neutral lipids [63]. These findings underline the important role of Wnt signaling in plasma lipid homeostasis and hepatic fat content [63,68,69,70]. Therefore, malfunction of the Wnt signaling pathway during adulthood is associated with cardiac diseases as divergent as myocardial infarction healing, hypertrophy, heart failure, arrhythmias, and atherosclerosis [59].

### 4.2. Cerebrovascular Disease

The cerebral vasculature is essential for brain development, activity, and homeostasis. It supplies the metabolic needs of this organ, which consumes one-fifth of the oxygen and one-quarter of the glucose used by the body [88]. 

The appearance and pattern of the nervous and vascular systems are closely coordinated, a process in which the Wnt pathway plays a particularly important role. In the brain, Wnt ligands activate a cell-specific surface receptor complex to induce intracellular signaling cascades that regulate neurogenesis, synaptogenesis, neuronal plasticity, synaptic plasticity, angiogenesis, vascular stabilization, and inflammation. The Wnt pathway is deregulated in trauma-mediated brain and vascular injuries.

Chronic hyperglycemia and microvascular disease contribute to cognitive dysfunction in both type 1 and T2DM, and both disorders are associated with mental and motor slowing and decrements of similar magnitude on measures of attention and executive functioning [89]. Furthermore, both types are characterized by neural slowing, increased cortical atrophy, microstructural abnormalities in white-matter tracts, and changes in concentrations of brain neurometabolites [89]. 

Additionally, there is an association between insulin resistance in T2DM patients and sympathetic overactivity, especially at night. This sympathetic overactivity is related to major vascular accidents [90]. Carotid IMT is increased in diabetes. It is an independent predictor of stroke, particularly of the ischemic subtype, and of stroke recurrence in both diabetic and nondiabetic populations [91]. A possible role of cIMT as a predictor of microangiopathy has also been suggested [91].

On the other hand, T2DM patients have an altered Wnt signaling pathway, and this may be the cause for the increased risk of these patients suffering a cerebrovascular accident, since this metabolic pathway is involved in the proper functioning of the cerebral vasculature (see Table 2).

Since β-catenin is a component of endothelial adherent junctions, its loss can reduce the stability of the junction and increase vascular permeability. Brain sections from hemorrhagic stroke patients show decreased β-catenin concentration, indicating defective β-catenin transcriptional activity in endothelial cells. Specifically, 60% of endothelial cells from hemorrhagic patients expressed a low level of nuclear β-catenin [76]. Xin-Wei et al. [71] observed that serum levels of sclerostin and Dkk-1 are markedly increased in patients with acute ischemic stroke. Furthermore, it was shown that Dkk-1 is induced in neurons located in the ischemic nuclei and the penumbra region after cerebral ischemia in animal models [92].

Chong et al. [75] showed that the loss of endogenous Wnt1 signaling is directly correlated with neuronal disappearance and functional deficit. Furthermore, transient overexpression of Wnt1 or the application of exogenous recombinant Wnt1 protein is necessary to preserve neurological function. Furthermore, Wnt3a attenuates neuronal apoptosis and improves neutral deficits after middle cerebral artery occlusion (MCAO) in rats [93]. In fact, intranasal administration of Wnt3a was shown to play an essential role in brain repair after ischemic stroke [78]

### 4.3. Peripheral Arterial Disease 

PAD generates tissue ischemia through arterial occlusions and insufficient collateral vessel formation [94]. Limb ischemia induced by obstructive arterial injury is aggravated by insufficient angiogenesis [94].

Many studies have demonstrated that the prevalence of PAD in patients with T2DM is higher than in nondiabetic patients [95]. Multiple metabolic aberrations in T2DM, such as advanced glycation end-products, low-density lipoprotein cholesterol, and abnormal oxidative stress, have been shown to worsen PAD [95].

PAD is common among diabetic patients with renal insufficiency, and most of the diabetic patients with end-stage renal disease (ESRD) have PAD [96]. ESRD is a strong risk factor for both ulceration and amputation in diabetic patients [96]. It increases the risk of nonhealing of ulcers and major amputation.

The Wnt/β-catenin signaling pathway is altered in T2DM patients, and it regulates endothelial inflammation, vascular calcification, and mesenchymal stem-cell differentiation, thus contributing to atherosclerosis disease [97,98]. In fact, Teng et al. [99] demonstrated that elevated serum sclerostin levels are a risk factor for PAD in the elderly. Sfrp5 inhibits Wnt signaling, thus regulating chronic inflammation and the development of atherosclerosis [100]. Wang et al. [82] demonstrated that Sfrp5 levels in serum and periarterial adipose tissue were significantly lower in PAOD patients than in control subjects. Wnt5a levels in serum and periarterial adipose tissue were significantly higher in PAOD patients than in control subjects. An error in the upregulation of Sfrp5 in PAOD can lead to rampant actions of Wnt5a that can result in metabolic and cardiovascular disorders [82].

## 5. Perspectives

Although the function of the Wnt pathway has traditionally been mainly associated with the regulation of osteoblastogenesis, scientific evidence from the last decade has shown that Canonical Wnt/β-catenin signaling is deeply involved in the development and maintenance of the vasculature and is closely linked to the pathogenesis of bone alterations and atherosclerotic processes [101]. This pathway is a promising target for the development of new protective and restorative therapeutic interventions. A full understanding of this signaling, through future studies, could be extremely important in the design of future innovative drugs capable of protecting both arterial walls and bone structure. However, targeting a complex multifactorial signal transduction pathway remains a major challenge.

Romosozumab is one of the most recent Wnt pathway modulator treatments. It is a humanized monoclonal antibody against sclerostin that is currently being used in clinical practice for the treatment of osteoporosis and high risk of fracture [102,103]. This suggests that the treatment of macro- and microvascular complications with drugs targeting components of the Wnt pathway is getting closer. 

Another way to target the Wnt pathway would be by influencing environmental factors such as nutrition or the microbiome. In this context, Wnt5a levels are lower in subjects with a high nutritional load of long-chain eicosatetraenoic acid, as well as in subjects with high α diversity of the gut microbiome. Modification of the nutrition habits that influence the microbiome could be an interesting strategy to prevent and/or treat systemic metabolic inflammation [64].

In fact, *Salvia miltiorrhiza* (frequently used herb in traditional Chinese medicine) exhibited antidiabetic activities when treating macro- and microvascular diseases in preclinical experiments and clinical trials through an improvement of redox homeostasis and inhibition of apoptosis and inflammation by regulating Wnt/β-catenin [13]. Other antidiabetic treatments targeting Wnt signaling pathway are aFGF for use in vascular complications of diabetes [12], glucagon-like peptide-1 (GLP-1) receptor agonists because of their effects on diabetic nephropathy or bone metabolism [14,104], and class α-glucosidase inhibitors and peroxisome proliferator-activated receptors (PPARs) agonists for use in Alzheimer disease (AD) [105].

The evidence shows that disturbances in Wnt signaling are crucial for the development of atherosclerosis; hence, restoring Wnt activity may help with reducing plaque burden. However, caution should be exercised with the use of Wnt pathway activators or inhibitors since an excessive activation or inhibition of the Wnt pathway may result in a deleterious effect. Thus, the use of compound targeting Wnt pathway should be carefully evaluated on an individual basis, an approach known as precision medicine, which has shown growing popularity [106].

## 6. Conclusions

In conclusion, the canonical Wnt signaling pathway has been traditionally associated with the regulation of bone remodeling as its main function. This relationship is so evident that there is currently a drug targeting this pathway to treat osteoporosis. However, with the deepening of knowledge thanks to research efforts, this pathway has been associated with many other extraskeletal functions. Although the vascular involvement of Wnt pathway until very recently has been considered silent, the weight of scientific evidence has currently positioned the Wnt signaling pathway as a key player in vascular pathophysiology. For this reason, many studies are currently focused on this pathway as a therapeutic target for diseases other than those related solely to bone. Although the results so far are still preliminary, in the near future, there will probably be several therapeutic strategies against macrovascular and microvascular complications targeting Wnt pathway through its different components.

## Figures and Tables

**Figure 1 ijms-23-06995-f001:**
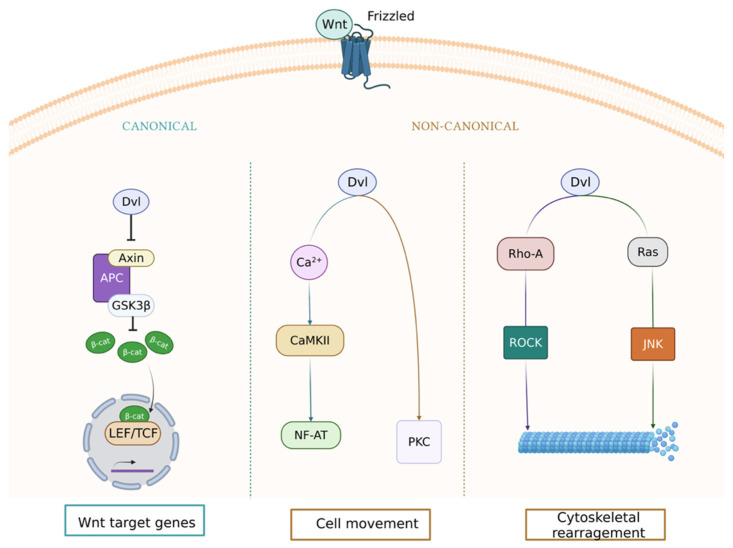
Canonical and noncanonical (Wingless-Int) Wnt signaling pathways. These pathways are activated when a Wnt ligand binds to the Frizzled (Fzd) receptor. The active canonical pathway is mediated by β-catenin, which translocates to the nucleus and acts as a coactivator of the transcription factor T-cell factor/lymphoid enhancer factor family (TCF/LEF), leading to upregulation of Wnt target genes. The two main noncanonical pathways are the Wnt/calcium and planar cell polarity (PCP) pathways. In the Wnt/calcium pathway, Wnt binding to Fzd activates Disheveled (Dvl), which stimulates calcium release from the endoplasmic reticulum, activating protein kinase C (PKC) and calmodulin-dependent kinase II (CamKII) and, in turn, the transcription factor nuclear factor of activated T cells (NFAT). The Wnt/PCP pathway is mediated by the Ras homolog family member A (RhoA) and Ras guanosine triphosphatases (GTPases) that lead to the activation of the RhoA/Rho-associated kinase (ROCK) or N-terminal kinase (JNK) axis. Created with BioRender.com.

**Figure 2 ijms-23-06995-f002:**
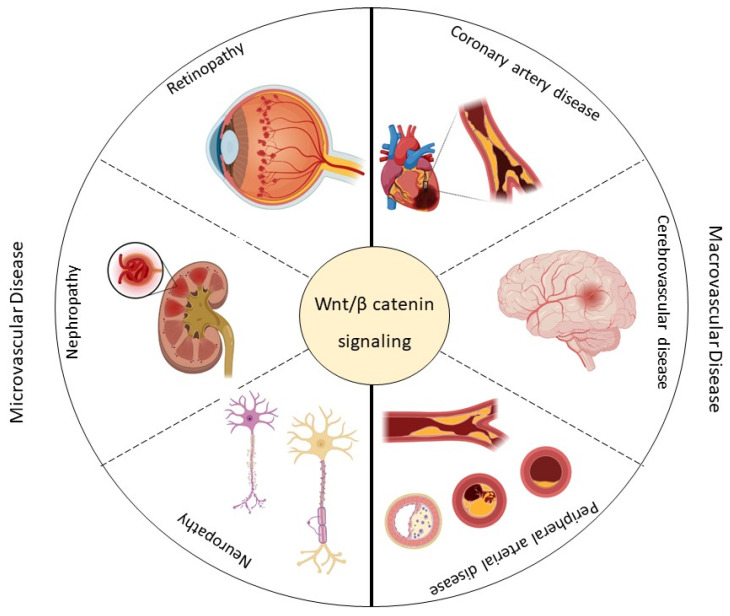
This scheme summarizes the micro- (retinopathy, nephropathy, and neuropathy) and macrovascular (coronary artery disease, cerebrovascular disease, and peripheral arterial disease) complications associated with type 2 diabetes mellitus (T2DM) and their possible association with the Wingless-Int (Wnt) signaling pathway. Created with BioRender.com.

**Table 1 ijms-23-06995-t001:** Components of the Wnt pathway that can be altered in the microvascular complications of T2DM ^1^.

Disease	Event	Component	Expression	In Vitro	In Vivo	Reference
Microvascular	Retinopathy	β-catenin	↑	Inflammation and angiogenesis	Retinal inflammation and vascular leakage	[26]
		LRP5/6	↑	Inflammation and angiogenesis	Retinal inflammation and vascular leakage	[26]
			↓	Lack of deeper retinal vessels	Significant decrease in pathological retinal neovascularization Significant decrease in retinal vascularization during development Affects blood–retinal barrier formation	[27]
		Dkk1	↑	Inhibition of the generation of reactive oxygen species (ROS)	Mitigated retinal inflammation and blocked overexpression of proinflammatory factors such as ICAM-1 and COX-2 Reduction in retinal vascular leakage and improvement of ischemia-induced retinal neovascularization	[26]
		Frizzled4	↑	Angiogenesis	Pathological neovascularization	[27]
		Dvl2	↓	Impaired angiogenesis	Significant decrease in pathological retinal neovascularization	[27]
		Claudin-5	↓	Significant suppression of endothelial cell sprouting	Suppression of pathological vascular growth and development	[27]
		Frizzled7	↑	Inflammation, angiogenesis, and oxidative stress	Pathological neovascularization	[28]
		SERPINA3K	↑	Inhibition of connective tissue growth factor overexpression	Antioxidation Anti-inflammatory Antifibrosis	[29]
		VLDLR	↑	Anti-angiogenesis Inhibited endothelial cell proliferation, migration, and tube formation	Improvement of ocular neovascularization,	[30]
		Endostatin	↑	Impaired angiogenesis	Reduced VEGF-induced retinal vascular permeability, neovascularization, and retinal detachment	[31]
		Kallistatin	↑	Anti-inflammation Anti-angiogenesis	Attenuation of ischemia-induced retinal neovascularization	[32]
		PEDF	↑	Anti-inflammation Anti-angiogenesis	Ameliorated retinal inflammation, vascular leakage, and neovascularization	[33]
		MiARN-184	↑	Anti-angiogenesis	Improves inflammatory responses, vascular leakage, and neovascularization.	[34]
	Nephropathy	β-catenin	↑	Reduced mesangial cell apoptosis Podocyte dysfunction	Glomerular albuminuria and subsequent glomerular injury	[35]
			↓	Mesangial cells apoptosis	Increased severity of streptozotocin-induced diabetes nephritis	[35]
		LEF1	↑	Enhanced proliferation and metastasis of renal cells	Renal cell carcinoma (RCC)	[36]
		LRP6	↓	Mesangial cell apoptosis	Attenuated renal inflammation, reduced proteinuria, and ameliorated fibrosis	[37]
		Wnt4	↑	Stimulation of mesenchymal-to-epithelial differentiation Podocyte dysfunction	Tubulo-interstitial fibrosis Glomerular albuminuria and subsequent glomerular injury	[35]
			↓	Mesangial cell apoptosis	Kidney tissue disorganization, as well as disease development and progression	[38]
		Dkk1	↑	Amelioration of podocyte apoptosis and viability	Restored podocyte function and decreased albuminuriaBone-mineral disorder syndrome	[35,39]
		TRPC6	↑	Podocyte injury	Excessive calcium influx in podocytes leading to foot process effacement, podocyte apoptosis, and subsequent glomerular damage	[35]
		Wnt9a	↑	Evoking of cell communication between senescent tubular cells and interstitial fibroblasts	Tubular senescence and renal fibrosis	[40]
		Wnt5a	↑	Increased ROS production	Mesangial cell apoptosis	[41]
		CTGF/CCN2	↑	LRP6 phosphorylation and accumulation of β-catenin	Attenuated renal inflammation, reduced proteinuria, and ameliorated fibrosis Mesangial cell apoptosis	[37]
		CTNNB1	↓	Improved podocyte motility	Damage to the basement membrane, albuminuria, and increased susceptibility to glomerular injury	[41]
		Wnt6	↓	Damaged tubulo-interstitium	Renal fibrosis	[42]
	Neuropathy	PORCN	↓	Slightly reduced expression of Wnt3a Significantly reduced expression of β-catenin, Dvl1, c-myc, GRP78, and MMP2 in the sciatic nerve	Decreased heat- and cold-induced hyperalgesia Increased motor nerve conduction speed Increased sensory nerve conduction speed Increased nerve blood flow Increased density of intraepidermal nerve fibers	[43]
		Dvl	↓	Significantly reduced expression of β-catenin, Dvl1, c-myc, GRP78, and MMP2 in the sciatic nerve	Decreased heat- and cold-induced hyperalgesia Increased motor nerve conduction speed Increased sensory nerve conduction speed Increased nerve blood flow Increased density of intraepidermal nerve fibers	[43]
		β-catenin	↓	Significantly reduced expression of β-catenin, Dvl1, c-myc, GRP78, and MMP2 in the sciatic nerve	Decreased heat- and cold-induced hyperalgesia Increased motor nerve conduction speed Increased sensory nerve conduction speed Increased nerve blood flow Increased density of intraepidermal nerve fibers	[43]
		Wnt3a	↑	Release of brain-derived neurotrophic factor in microglial cells	Allodynia	[44]
		XAV939	↑	-	Effective attenuation of neuropathic pain induction Drastic attenuation of the development of allodynia	[44]

^1^ LRP: LDL receptor-related protein; Dkk: Dickkopf; ROS: reactive oxygen species; ICAM: intercellular adhesion molecule; COX: cyclooxygenase; Fzd: Frizzled; Dvl: Disheveled; VLDLR: very-low-density lipoprotein receptor; VEGF: vascular endothelial growth factor; PEDF: pigment epithelium-derived factor; LEF: lymphoid enhancer factor; Wnt: Wingless-Int; TRPC: transient receptor potential canonical; CTGF/CCN2: connective-tissue growth factor; CTNNB: β-catenin gene; PORCN: Porcupine; GRP78: glucose-regulated protein; MMP: matrix metalloproteinase; ↑: upregulation; ↓: downregulation.

**Table 2 ijms-23-06995-t002:** Components of the Wnt pathway that can be altered in the macrovascular complications of T2DM ^2^.

Disease	Event	Component	Expression	In Vitro	In Vivo	Reference
Macrovascular	Coronary artery disease	Scl	↑	Endothelial dysfunction, alteration on proliferation, and migration of vascular smooth muscle cells	Atherosclerotic process, abnormal intima-media thickness, carotid plaques, aortic calcifications, and mortality	[59,60]
		Dkk-1	↑	Regulates platelet-mediated inflammation and contributes to plaque de-escalation	Ischemic stroke and cardiovascular death	[61]
			↑	Endothelial activation and release of inflammatory cytokines Endothelial–mesenchymal transition in aortic endothelial cells	Onset and progression of atherosclerosis	[62]
		LRP6	↓	LDL uptake was significantly lower in lymphoblastoid cells	Elevated plasma cholesterol and elevated plasma LDL, triglyceride, and fatty liver levels	[63]
		Wnt5a	↑	Induction of inflammatory gene expression GM-CSF, IL-1a, IL-3, IL-5, IL-6, IL-7, IL-8, CCL2, CCL8, and COX-2 in human aortic endothelial cells	Elevation of triglyceride levels, vascular insulin resistance, and endothelial dysfunction	[64]
			↑	Macrophage activation	Increased recruitment of inflammatory cells and amplified inflammatory response	[65]
		Dkk-3	↓	Increased intima-media thickness of the carotid artery	Delayed reendothelialization and aggravated neointima formation	[66]
			↑	Induces differentiation of vascular progenitors and fibroblasts into smooth muscle cells	Larger and more vulnerable atherosclerotic lesions with more macrophages, fewer smooth muscle cells, and less extracellular matrix deposition	[67]
		TCF7L2	↓	Loss of differentiation of vascular smooth muscle cells	Medial aortic hyperplasia	[68]
		Wnt2	↑	Regulates smooth muscle cell migration	Triggers intima-media thickening	[69]
		LRP5	↓	Activation of proinflammatory genes (interferon γ, IL15, IL18, and TNF ligand superfamily 13b).	Larger aortic atherosclerotic lesions	[70]
	Cerebrovascular disease	Scl	↑	Arterial calcification	Ischemic stroke caused by atherosclerotic stroke of large arteries or occlusion of small arteries	[71]
		Dkk1	↑	Biomarker for the presence of coronary atherosclerotic plaque	Carotid atherosclerosis, stable angina, and myocardial infarction Poor prognosis 1 year after ischemic stroke	[72]
		miR-150-5p	↑	Regulates the Wnt signaling pathway and participates in cell proliferation and apoptosis by downregulating p53	Inhibition of cell proliferation, colony formation, and tumor growth	[73]
			↓	CD133^−^ cells acquire a stem-cell-like phenotype	>Glioma	[74]
		β-catenin	↑	Key regulators for cadherin-mediated cell–cell adhesion	Glioma Higher degree of malignancy of the tumor	[74]
		Wnt1	↓	Neuronal disappearance and increasing functional deficits	Oxidant stress and cerebral ischemia	[75]
		claudin-1	↓	Neuronal damage	Increased permeability of the blood–brain barrier, petechial hemorrhage in the brain, neuronal injury, and central nervous system inflammation	[76]
		Claudin-3	↓	Neuronal damage	Intracerebral petechial hemorrhages	[77]
		Wnt3a	↑	Alleviates neuronal apoptosis at the cellular and subcellular levels	Neuroprotection in traumatic brain injury, and ischemic stroke	[78]
		LRP6	↓	Increased expression of inflammatory genes after middle artery occlusion	Risk of ischemic stroke, larger heart attack, and severe motor deficits	[79]
		Wnt5	↑	Enhanced endothelial activation type 1 inflammatory mediator to promote endothelial activation type 2	Brain aging Inflamed atheroma plaques	[80]
		miRNA-148b	↓	Attenuates neural stem-cell proliferation and differentiation	Reduces ischemic injury and improves neurological function	[81]
	Peripheral arterial disease	Wnt5a	↑	Endothelial dysfunction	Increased risk of peripheral arterial occlusive disease, as well as metabolic and cardiovascular disorders	[82]
		Sfrp5	↓	Inhibition of cardiac fibroblast proliferation and migration Inflammation and myocardial injury	ST-segment elevation myocardial infarction, metabolic syndrome, and increased risk of peripheral arterial occlusive disease	[82]
		CTHRC1	↑	Synovial hyperplasia, contributes to the inflammatory microenvironment, and promotes pannus invasion through increased motility and invasion of synoviocytes	Increased risk of systemic lupus erythematosus, development of rheumatoid arthritis, and severity of the disease	[83]
		ALKBH5	↑	Reduced proliferation and migration and decreased viability in hypoxic cardiac microvascular endothelial cells	Impaired hypoxic tube formation, but not the normoxic cardiac microvascular endothelial cells	[84]

^2^ Dkk: Dickkopf; LRP: LDL receptor-related protein; LDL: low-density lipoprotein; Wnt: Wingless-Int; GM-CSF: granulocyte macrophage-colony stimulating factor; IL: interleukin; CCL: collagen crosslinking; COX: cyclooxygenase; TCF: transcription factor; TNF: tumor necrosis factor; Scl: sclerostin; CD: cluster of differentiation; Sfrp: secreted Frizzled-related proteins; CTHRC: collagen triple helix; ↑: upregulation; ↓: downregulation.

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
