# Peer review of "The Contribution of Wnt Signaling to Vascular Complications in Type 2 Diabetes Mellitus"

_ijms, 2022, doi:10.3390/ijms23136995_

Round 1

Reviewer 1 Report

Here, Sanabria-de la Torre and colleagues reviewed the involvement of the canonical Wnt signaling in the pathogenesis of micro- and macrovascular complications of type 2 diabetes. The manuscript is well organized and provides an updated and comprehensive overview of the topic.

The authors should address the following comments:

- While the role of the Wnt signaling in osteogenesis is briefly depicted, it would be useful to add a paragraph on its involvement in endothelial cell biology, beyond T2DM and its complications

- The authors should briefly discuss the involvement of the Wnt pathway in modulating or mediating the effects of antidiabetic drugs (see PMID 32715762, 32615283 etc.)

- In line 198, could the authors comment more on the link between Wnt and cellular senescence?

- In table 1, information and reference on LEF1 upregulation are missing.

Author Response

RESPONSE TO REVIEWER #1

First, we would like to thank for your effort in reviewing our manuscript and for your constructive comments, which have undoubtedly contributed to improve the quality of our manuscript.

Here, Sanabria-de la Torre and colleagues reviewed the involvement of the canonical Wnt signaling in the pathogenesis of micro- and macrovascular complications of type 2 diabetes. The manuscript is well organized and provides an updated and comprehensive overview of the topic.

We are very grateful for your feedback on our work.

- While the role of the Wnt signaling in osteogenesis is briefly depicted, it would be useful to add a paragraph on its involvement in endothelial cell biology, beyond T2DM and its complications.

As the reviewer suggests, we have added a paragraph on Introduction section with the following information:

“Also, the angiogenic activity of endothelial cells (ECs) is influenced by Wnt signaling. Endothelial dysfunction is the earliest and most fundamental pathological change in diabetes. Wnt, Frizzled (Fzd) and follistatin-related protein (FRP) genes are expressed by ECs and VSMCs; β-catenin is stabilized in neovascular endothelium and neointimal smooth muscle. Activation of the Wnt signaling pathway can lead to vessel remodelling, while inhibition of Wnt signaling can lead to vessel regression.”

- The authors should briefly discuss the involvement of the Wnt pathway in modulating or mediating the effects of antidiabetic drugs (see PMID 32715762, 32615283 etc.)

We agree with the reviewer comment. In this respect, we have added references 12, 14, 104 and 105 to comment on this matter:

  • Sun J, Huang X, Niu C, Wang X, Li W, Liu M, Wang Y, Huang S, Chen X, Li X, Wang Y, Jin L, Xiao J, Cong W. aFGF alleviates diabetic endothelial dysfunction by decreasing oxidative stress via Wnt/β-catenin-mediated upregulation of HXK2. Redox Biol. 2021 Feb;39:101811.
  • Huang, L.; Lin, T.; Shi, M.; Chen, X.; Wu, P. Liraglutide suppresses production of extracellular matrix proteins and ameliorates renal injury of diabetic nephropathy by enhancing Wnt/β-catenin signaling. J. Physiol. Renal Physiol.2020, 319, F458–F468, doi:10.1152/AJPRENAL.00128.2020.
  • Li, Y.; Fu, H.; Wang, H.; Luo, S.; Wang, L.; Chen, J.; Lu, H. GLP-1 promotes osteogenic differentiation of human ADSCs via the Wnt/GSK-3β/β-catenin pathway. Cell. Endocrinol.2020, 515, doi:10.1016/J.MCE.2020.110921.
  • Manandhar, S.; Priya, K.; Mehta, C.H.; Nayak, U.Y.; Kabekkodu, S.P.; Pai, K.S.R. Repositioning of antidiabetic drugs for Alzheimer’s disease: possibility of Wnt signaling modulation by targeting LRP6 an in silico based study. Biomol. Struct. Dyn.2021, doi:10.1080/07391102.2021.1930583.

We have included this information in different sections of the manuscript, where we consider it is appropriate:

In section 1.Wnt signaling pathway in the vasculature:

“In fact, the activation of Wnt/β-catenin signaling pathway by the administration of aFGF alleviates diabetic endothelial dysfunction. Recombinant human aFGF would be an effective treatment of diabetic vascular complications due to its intervention in the Wnt pathway.”

“Agreeing with these evidences, several drugs targeting Wnt signaling have been shown to have a positive effect in the treatment of some vascular disorders. For example, Salvia miltiorrhiza; recombinant human aFGF and liraglutide.’’

In section 3.2. Diabetic Kidney Disease:

“For example, liraglutide treatment ameliorates renal injury of diabetic nephropathy by enhancing Wnt/β-catenin signaling [49].”

In section 5. Perspectives

“Other antidiabetic treatments targeting Wnt signaling pathway are aFGF for use in vascular complications of diabetes; glucagon-like peptide-1 (GLP-1) receptor agonists because of its effects on diabetic nephropathy or bone metabolism; class α-glucosidase inhibitors and peroxisome proliferator-activated receptors (PPARs) agonists for use in Alzheimer Disease (AD).”

- In line 198, could the authors comment more on the link between Wnt and cellular senescence?

The link between the Wnt signaling pathway and cellular senescence is further explored in section 3.2. Diabetic kidney disease (the text is marked in blue in the reviewed manuscript). We have focused on cellular senescence associated with diabetic nephropathy since there is evidence that this signaling pathway is involved in the aging of cell types associated with renal diseases, as discussed in the manuscript. However, apart from this, there is no more extensive evidence to discuss the involvement of Wnt signaling pathway in cellular senescence in the scientific literature.

“In diabetic status, the accumulated intracellular reactive oxygen species (ROS) might divert the limited pool of β-catenin from TCF/LEF to forkhead box O- (FOXO-) mediated transcription that leads to insulin deregulation [44]. Recent evidence suggests that the renin-angiotensin-aldosterone system (RAS) is mediated by Wnt/β-catenin signaling [38]. Ectopic β-catenin causes the upregulation of all RAS genes [38]. Notably, RAS activation contributes to renal aging through various mechanisms. Several studies discover that angiotensin II can induce senescence in renal cells[45]. Angiotensin II induces premature senescence via both signal transducer and activator of transcription 3 (STAT3) and mammalian target of rapamycin (mTOR)-regulated autophagy and the p53/p21 pathway [45], which further drives fibrosis and redox state [46]. Furthermore, recent studies reported that RAS activation plays a role in Wnt9a-induced tubular senescence and renal fibrosis [47] (Table 1).”

So, components of Wnt pathway are potential therapeutic targets of cellular senescence in diabetic nephropathy and provide important clues for clinical strategies (Table 1).”

- In table 1, information and reference on LEF1 upregulation are missing.

According to the reviewer comment, LEF1 information in Table 1 has been updated. We have added reference 36:

  • Liu, Y.; Shang, D. Transforming growth factor-β1 enhances proliferative and metastatic potential by up-regulating lymphoid enhancer-binding factor 1/integrin αMβ2 in human renal cell carcinoma. Cell. Biochem.2020, 465, 165–174, doi:10.1007/S11010-019-03676-8.

All the changes have been marked in red font in the text.

Thank you very much for taking the time and effort necessary to review our manuscript.

Reviewer 2 Report

This is a well balanced and thorough review on Wnt signalling in T2DM.

I would suggest two minor typo amendments.

Line 199. Pluralize “mechanism” in “mechanisms”.

Like 389. Amend “effects” in “effect”.

Author Response

RESPONSE TO REVIEWER #2

First, we would like to thank for your effort in reviewing our manuscript and for your constructive comments, which have undoubtedly contributed to improve the quality of our manuscript.
We have corrected the minor typo amendments suggested in the manuscript.
Line 199. Pluralize “mechanism” in “mechanisms”.
In the reviewed manuscript, “mechanism” has been changed by “mechanisms”
Like 389. Amend “effects” in “effect”.
In the reviewed manuscript, “effects” has been changed by “effect”.
The changes have been marked in red font in the text.
Thank you very much for taking the time and effort necessary to review our manuscript.